# Hybrid Approach to Identifying Druglikeness Leading Compounds against COVID-19 3CL Protease

**DOI:** 10.3390/ph15111333

**Published:** 2022-10-28

**Authors:** Imra Aqeel, Muhammad Bilal, Abdul Majid, Tuba Majid

**Affiliations:** 1Biomedical Informatics Research Lab, Department of Computer & Information Sciences, Pakistan Institute of Engineering & Applied Sciences, Nilore, Islamabad 45650, Pakistan; 2Experimental Continuum Mechanics Research Group, Department of Mechanical and Process Engineering, ETH Zurich, 8092 Zürich, Switzerland

**Keywords:** SARS-CoV-2, 3C-like protease, drug repurposing, regression model, bioactive molecules, molecular docking

## Abstract

SARS-CoV-2 is a positive single-strand RNA-based macromolecule that has caused the death of more than 6.3 million people since June 2022. Moreover, by disturbing global supply chains through lockdowns, the virus has indirectly caused devastating damage to the global economy. It is vital to design and develop drugs for this virus and its various variants. In this paper, we developed an in silico study-based hybrid framework to repurpose existing therapeutic agents in finding drug-like bioactive molecules that would cure COVID-19. In the first step, a total of 133 drug-likeness bioactive molecules are retrieved from the ChEMBL database against SARS coronavirus 3CL Protease. Based on the standard IC50, the dataset is divided into three classes: active, inactive, and intermediate. Our comparative analysis demonstrated that the proposed Extra Tree Regressor (ETR)-based QSAR model has improved prediction results related to the bioactivity of chemical compounds as compared to Gradient Boosting-, XGBoost-, Support Vector-, Decision Tree-, and Random Forest-based regressor models. ADMET analysis is carried out to identify thirteen bioactive molecules with the ChEMBL IDs 187460, 190743, 222234, 222628, 222735, 222769, 222840, 222893, 225515, 358279, 363535, 365134, and 426898. These molecules are highly suitable drug candidates for SARS-CoV-2 3CL Protease. In the next step, the efficacy of the bioactive molecules is computed in terms of binding affinity using molecular docking, and then six bioactive molecules are shortlisted, with the ChEMBL IDs 187460, 222769, 225515, 358279, 363535, and 365134. These molecules can be suitable drug candidates for SARS-CoV-2. It is anticipated that the pharmacologist and/or drug manufacturer would further investigate these six molecules to find suitable drug candidates for SARS-CoV-2. They can adopt these promising compounds for their downstream drug development stages.

## 1. Introduction

Novel coronavirus (nCoV-19) is a rapidly spreading pandemic. The International Committee on Taxonomy of Viruses (ICTV) officially named severe acute respiratory syndrome coronavirus 2 (SARS-CoV-2) on February 11, 2020 [1]. At first, coronavirus-2 appeared in December 2019 in Asia and then spread out worldwide. A total of 228 countries and more than 500 million people got infected. SARS-CoV-2 is like MERS-CoV and SARS-CoV. Both these viruses have caused severe acute respiratory syndrome. There are seven strains of Alpha and Beta coronaviruses in human coronaviruses. HCoV-229E and HCoV-NL63 belong to the type of alpha-coronaviruses. On the other hand, HCoV-HKU1, HCoV-OC43, SARS-CoV, MERS-CoV, and SARS-CoV-2 belong to beta-coronaviruses [2]. COVID-19 virus is a single-strand ribonucleic acid (ssRNA) virus that consists of multiple structural and non-structural proteins. The structural proteins have four different types: spike (S), membrane (M), envelope (E), and nucleocapsid (N) proteins. However, non-structural proteins contain sixteen different types, named NSP1, NSP2, NSP3…, and NSP16. These proteins are mainly more responsible for spreading out SARS-CoV-2 than other types of proteins. Consequently, these proteins are considered potential targets to prevent SARS-CoV-2, especially the 3C-like protease (3CL^pro^ or M^pro^), which is crucial for replication [3]. Figure 1a shows a visual model of SARS-CoV-2 with all the constituent proteins, and Figure 1b depicts its large genome size of 29.9 kb, starting from 5ʹ to 3ʹ. This virus has the inherent capability of auto-reproduction into sixteen different types of non-structural proteins.

Upon entrance into the host cell, the viral genome is translated to produce two overlying polyproteins named *pp1a* and *pp1b* [4]. During the proteolytic activity, these polyproteins are excised from the 3CL protease (3CL^pro^, also known as the Main protease (M^pro^)). These proteins work with a papain-like protease to slice the polyproteins to produce a total of sixteen functional nonstructural proteins (NSPs). It was reported that the eleven slicing sites of polyprotein 1ab were shared and operated by only the 3CL^pro^ of SARS, and no other human protease was involved in the slicing process [4]. To initiate viral replication, the viral replication transcription complex (RTC) is assembled by the sliced NSPs.

The computational drug discovery process has become a crucial strategy to develop the drug against COVID-19. It can be an effective tool to save money and reduce the time for drug discovery/repurposing [5]. Recently, machine learning (ML) approaches have been employed for data modeling and drug discovery applications. Various online medical databases that contain sufficient information related to bioactive molecules are available. This has made it possible to employ the ML approaches-based QSAR model to quickly develop vaccines for the COVID-19 pandemic [6]. Due to stringent storage requirements, this vaccine is rather difficult to transport and warehouse. Moreover, successful virus vaccinations for humans and animals are seriously hampered by vaccine-associated increased illness [7]. This has shown that people are not as receptive to getting vaccinated as they are to taking drugs [8]. On the other hand, underdeveloped countries suffered the most from the pandemic, with the official death tolls of India and Brazil, at the time of writing this manuscript, being 525,000 and 672,000, respectively. Since the start of this pandemic, in India and Indonesia, millions of people have been pushed back into poverty [8]. Such nations would benefit from cheap, easy-to-store, and rapidly deployable effective drugs against SARS-CoV-2.

In this study, we repurposed existing therapeutic agents by examining drug-like bioactive molecules for COVID-19. For this purpose, we have developed a hybrid approach that combines the useful information extracted via various bioinformatics tools such as SWISSADME, AUTODOCK VINA, and PYMOL. In the proposed approach, a total of 133 bioactive compounds are retrieved from a large chemical space at the ChEMBL database against the SARS-CoV 3CL protease target. The EDA analysis is carried out using a molecular descriptor via Lipinski’s rule. In EDA analysis, the Mann–Whitney U test determined the significant difference between fifteen bioactive molecules belonging to the active class and the bioactive molecules of the inactive class. The PubChem fingerprints are computed using RDKit for the balanced dataset. In the next step, QSAR modeling is carried out using six diverse types of regression algorithms: Extra Tree, Gradient Boosting, XGBoost, Support Vector, Decision Tree, and Random Forest. These algorithms are being employed in the cheminformatics literature for drug discovery. These models predict the molecules with the best biological activities. Our comparative analysis demonstrated that the Extra Tree Regressor (ETR)-based QSAR model has improved prediction results related to the bioactivity of chemical compounds as compared to other ML-based QSAR models. In this study, from ADMET analysis, we have identified thirteen novel bioactive molecules for SARS-CoV-2. The efficacy of these bioactive molecules is computed in terms of binding affinity using molecular docking, and then the six most favorable potential drug candidates are short-listed, with the ChEMBL IDs 187460, 222769, 225515, 358279, 363535, and 365134.

The rest of the paper is arranged as follows: a literature review of recent methods used to identify lead compounds is given below. The results and discussions are presented in Section 2. In Section 3, the material and proposed hybrid framework is explained. Section 4 highlights the conclusion of this study.

### Literature Review

In recent years, various ML algorithms have been proposed in the development of vaccines and drug development processes. In particular, several efforts are carried out to perform the virtual screening of bioactive molecules that could inhibit SARS-CoV-2. In a study [9], a unique drug similarity model was developed using the characteristics of existing drugs such as remdesivir, dexamethasone, and baricitinib to inhibit COVID-19. The interactive compounds were retrieved using the known chemical–chemical interaction to repurpose existing drugs against SARS-CoV-2. A two-tier clustering approach was developed, in which tier-1 used the t-Distributed Stochastic Neighbor Embedding (t-SNE) and tier-2 analyzed the two-cluster analysis. Then, molecular docking was performed to check the validation of the top drug candidates. In a research work [10], a network-based model was developed to explore the drug candidates against COVID-19. The genome similarity was used among SARS-CoV-2 and other viruses such as SARS and MERS. A molecular network was designed and found 30 suitable drugs including chloroquine, thalidomide, and rographolide. In another work [11], the authors performed virtual screening to repurpose the drugs against COVID-19. They identified the two existing drugs as Lurasidone and Talampicillin. They also identified two drug-likeness molecules from the Zinc database. In the work, they performed molecular dynamic simulation, and ADMET analysis was also carried out.

In another research work [12], a deep learning approach was employed using SVM, logistic regression, and random forest to calculate the molecular descriptor. From QSAR modeling, they calculated the binding affinities of proteins with the drug target. In [13], quantitative high-throughput screening (qHTS) was performed to investigate the potential inhibitors against SARS-CoV-2 3CL Protease. In another work [14], the uses and limitations of bioinformatics tools was explained to prevent and reduce the spread of SARS-CoV-2. In [15], the authors proposed that niacin would be a potential therapy for COVID-19. They explored the properties of CRC patients and investigated the prognosis, biological functions, survival rate, and binding capacity. Further, in [16], some FDA-approved drug candidates were proposed for the therapy of COVID-19. The Jaccard similarity analysis was carried out on the lung cancer drugs dataset taken from Drug Bank and PubChem using graph neural network models.

A method of virtual screening was developed and found multiple drug candidates [17]. In that work, two antiviral candidates of bafetinib and 7-hydroxystaurosporine gave better results. In another work [18], the authors proposed shape similarity-based pre-docking and interaction similarity-based post-docking methods to screen the drugs against COVID-19. In [19], the authors analyzed the literature-based discovery methods to reposition the drugs for COVID-19. They also compared literature-based methods such as BITOLA, Arrowsmith, and SemBT. In another study [20], a graphical neural network based on the learning of the embedding of chemical compounds to predict molecular properties for COVID therapy was suggested. In another research work [21], the authors investigated various ML models and deep learning methods to find the chemical compounds against COVID-19 through the Ligand-Based Drug Designing approach. However, in another study [22], the authors employed the Naïve Bayes ML algorithm and Drug Bank to screen the anti-COVID compound. In an interesting study [23], the authors investigated the effectiveness of antiviral medications against COVID-19 using a plaque reduction assay.

## 2. Results and Discussion

In this section, we will discuss the exploratory data analysis, evaluation of the proposed model, comparative analysis, ADMET analysis, and molecular docking, respectively.

### 2.1. Exploratory Data Analysis

The exploratory data analysis (EDA) is performed using Lipinski’s rule-of-five descriptor. The chemical space of the descriptor shows the structure–activity relationship. The bioactive compounds, based on the IC50 value, were categorized as active, inactive, and intermediate classes. Its detail is given in Section 3.1.3.

Statistical analysis is used to find the significant difference between both the active and inactive classes. For this purpose, the Mann–Whitney U test is employed. This nonparametric test determines whether the dependent variable differs between two independent groups. It evaluates if the dependent variable’s distribution is the same for the two groups and, consequently, comes from the same population. Table 1 illustrates the Mann–Whitney U test results regarding significant differences in both bioactivity classes.

This table indicates that both classes are different. The interpretation of four descriptors—MW, NumHacceptors, NumHDonors, and pIC50—highlights that both classes are significantly different, except for the logP descriptor. This descriptor shows no significant difference between the two classes. The LogP values for the active class are the lowest, while the other class’ differences are minuscule. Hence, for logP, the test gave no significant difference.

Figure 2a–e represent the box plots of the Mann–Whitney U test results, in which LogP, MW, NumHAcceptors, NumHDonors, and pIC50 are shown in Figure 2a, Figure 2b, Figure 2c, Figure 2d, and Figure 2e, respectively. LogP is a commonly utilized metric for figuring out a compound’s lipophilicity, as well as its permeability and penetration of membranes. However, the molecular weight (MW) of a substance is very important to estimate the right size of a compound. Its numerical values are crucial for transit through a lipid membrane. On the other hand, NumHAcceptors and NumHDonors are used to measure the hydrogen bonding capacity and refer to the quantity of hydrogen bond acceptors and donors, respectively.

According to the analysis of the box plots, the margins of the boxes indicated that there was a negligible difference between both bioactivity classes for LogP. However, the MW, NumHAcceptors, NumHDonors, and pIC50 box plots revealed a clear difference between both classes.

First, based on their IC50 values, we categorized whether the bioactive molecule belongs to the active, inactive, or intermediate class. Then, we applied the statistical test of the Mann–Whitney U test to determine the significant difference between the active and inactive classes. At the outcome of this analysis, we found that 15 bioactive molecules belonging to the active class and 58 molecules belonging to the inactive class are significantly different. The fifteen bioactive molecules of the active class, with their ChEMBL IDs, chemical formulae, PubChem IDs, Isomeric SMILES, and 3D structures, are tabulated in Table 2.

### 2.2. Evaluation of the Proposed Model

The proposed QSAR model is developed using the ETR algorithm for 76 bioactive molecules in the balanced dataset. For this purpose, X and Y data matrices are prepared, in which 881 PubChem fingerprints are placed in the X matrix, and their corresponding pIC50 values are placed in the Y matrix. To train the model, the input dataset is split into a 70/30 training-to-testing ratio. Here, 70 represents the dataset used to train the model, and 30 represents the dataset used to test the model. Now, the QSAR Model is evaluated using two well-known performance measures: coefficient of determination (R^2^) and root mean square error (RMSE). The value of R^2^ highlights the statistical metric of fit that measures how much variation of a dependent variable (pIC50) is explained by the independent features/variables. Its value ranges from 0 to 1. The higher the value of R^2^, the better the model would be. However, RMSE represents the relative error between the experimental and model-predicted values of pIC50. Figure 3 demonstrates the experimental and ETR model-predicted pIC50 for the training and testing data.

Most commonly, in QSAR modeling, the performance is measured in terms of the difference between the values of R^2^ and Q^2^. This difference should be less than 0.3 [24]. Moreover, a value of Q^2^ greater than 0.5 shows the good regression performance of the model, and a value above 0.9 shows an excellent performance. Our model has obtained an R^2^ value of 0.63 for the training data and a Q^2^ value of 0.73 for the testing data. There is a minor difference (0.10) between R^2^ and Q^2^. This indicates that the proposed QSAR prediction model is the most suitable, having a sufficient estimation power of pIC50.

### 2.3. Comparative Analysis

The performance of the proposed QSAR model is compared with five other state-of-the-art models. These prediction models are trained, and their performances are evaluated using two performance measures of R^2^ and RMSE for testing the dataset. The graphical performance comparison among various regression models in terms of experimental and predicted pIC50 values is shown in Figure 4a–f. The highest value of R^2^ = 0.73, for the testing data, highlights a strong relationship between the experimental and predicted pIC50 values. Figure 4b shows the relationship between the experimental pIC50 and predicted pIC50 values of GradientBoosting Regressor (GBR), and the corresponding value of R^2^ is 0.62. However, Figure 4c shows an R^2^ value of 0.59 for XGBoost Regressor (XGBR). Figure 4d–f show a relatively lower fit of 0.59, 0.58, and 0.52 for Support Vector Regressor (SVR), Decision Tree Repressor (DTR), and Random Forest Repressor (RFR), respectively.

Table 3 highlights the performance comparison of various regression models in terms of R^2^, Mean Squared Error (MSE), and RMSE values. A smaller RMSE value indicates that the model predicts the data accurately. This table indicates that the lowest RMSE value is 0.074 in our ETR model. The GBR model has a higher RMSE value of 0.078 than our proposed model. Similarly, there are higher RMSE values of 0.089, 0.078, 0.092, and 0.089 for XGBR, SVR, DTR, and RFR, respectively.

### 2.4. ADMET Analysis

The ADMET analysis of the bioactive molecules belonging to the active class is carried out in terms of six main categories: physicochemical properties, lipophilicity, water solubility, pharmacokinetics, drug-likeness, and medicinal chemistry. There are ten measures in physicochemical properties, five measures in lipophilicity, three measures in water solubility, nine measures in pharmacokinetics, six measures in drug-likeness, and four measures in medicinal chemistry. In the study for the ADMET analysis, we have computed a total of 37 properties related to absorption, distribution, metabolism, excretion, and toxicity.

The physicochemical properties are expressed in terms of MW, Topological Polar Surface Area (TPSA), Number of Hydrogen Bonds Acceptors (NHA), and Number of Hydrogen Bonds Donors (NHD). NHA and NHD, relevant to the polarity, are the main measures of the drug-like molecules. The drug needs to be relatively non-polar to pass through most membranes. A drug needs to be polar to be water-soluble. Overly nonpolar drugs may not be water-soluble, or they may attach to dietary ingredients or blood proteins too firmly. These values related to physicochemical properties are shown in column 2 of Table 4. Further, the values of lipophilicity are calculated in terms of consensus log P_o/w_. Its value is computed by taking an average of the values of ilogp, xlogp3, wlogp, mlogp, and logp (silicos-IT). These values are given in column 3 of Table 4. The water solubility is categorized into five classes: poorly soluble, soluble, moderately soluble, highly soluble, and very soluble. These values are given in column 4 of Table 4.

The Pharmacokinetics properties are expressed in terms of the three most significant measures: gastrointestinal (GI) absorption, Blood–Brain Barrier (BBB) permeant, and skin permeation (log Kp). A high value of GI absorption means that the drug is absorbable, and a low values means that it is not absorbable. However, the binary value (yes) of the BBB measure indicates that bioactive molecules can penetrate through BBB, and the binary value (no) shows that they cannot penetrate through BBB. On the other hand, the value of skin permeability (Kp) serves as a proxy for the number of molecules absorbed in the skin; a more negative value indicates lesser skin absorption. These values related to pharmacokinetics properties are computed and given in column 5 of Table 4.

The binary values of drug-likeness properties are expressed as “yes/no”. A “yes” value is computed if 3 of 5 rules are satisfied according to the drug-likeness criteria, such as Lipinski, Ghose, Veber, Egan, and Muegge rules; otherwise, it is assigned “no”. These values are given in column 6 of Table 4. The property of medicinal chemistry is measured in terms of synthetic accessibility. Its numerical score for drug-like molecules is in the range of 1–10. A numerical value of one indicates that it is very easy to synthesize, and a value of ten indicates that it is very difficult to synthesize for chemist guidance. This numerical value is given in column 7 of Table 4.

The most important characteristics, which have been taken into account in most of the metrics used to create limits in the drug-like chemical space, are lipophilicity, molecular size, and polarity [25]. There is substantial proof that drugs with a higher lipophilicity and molecular weight, such as those with a high molecular corpulence, are more likely to be dropped during clinical trials. These are linked to complications with oral absorption. For a drug-like compound, the numerical values of MW should be less than 480 g/mol. This table shows that ChEMBL ID 212454 has a relatively higher value of MW 585.19 g/mol than the stated criteria. All physicochemical properties of this compound, such as TPSA, NHA, and NHD, are satisfied, except for the numerical value of MW. The MW measure is the most important physicochemical property. Further, this compound also has low GI absorption, is poorly soluble, and violates three of the five drug-likeness criteria. Therefore, this compound cannot be a potential drug candidate.

On the other hand, ChEMBL ID 212218 has a value of TPSA of 136.16 Å^2^. That is greater than the standard criterion TPSA value of 130 Å^2^. It cannot penetrate through the blood–brain barrier, and it has poor penetration through the cell membrane. Further, this compound has also low GI absorption. Therefore, this compound does not satisfy pharmacokinetic properties as well. We infer that this compound cannot be a potential drug candidate.

The other thirteen compounds that fulfill the ADMET criteria can be potential drug candidates. For further validation, these thirteen compounds were investigated for the molecular docking process.

### 2.5. Molecular Docking

The filtered molecules are obtained from ADMET analyses that belong to the active class. In the final stage of the current study, molecular docking is performed, and the binding affinity of the selected inhibitors is checked with the target protein 7JSU. The value of binding affinity is represented in the unit of Kcal/mol. In the ligand-based docking approach, we want to investigate the efficacy of the selected inhibitors. Figure 5 visually depicts the best pose of six bioactive molecules towards the target protein that corresponds to the lowest binding affinity in the range of −8.4 to −7.0. The best ligands pose towards the target protein corresponds to the most negative binding energy value.

Table 5 demonstrates the results of the binding affinities of thirteen selected ligands’ computed molecular docking towards the target protein 7JSU. A lower RMSD measure indicates a higher correctness in the docking geometry of the ligand molecule from its reference position in the original protein complex. In the study, we found all thirteen selected ligands that have an RMSD value of zero at their best poses.

In general, a lower value of binding affinity indicates a stronger interaction between the protein and ligand. From the results in Table 5, we have selected six bioactive molecules, with the ChEMBL IDs 187460, 222769, 225515, 358279, 363535, and 365134, that possess the lowest binding affinity in the range of −8.4 to −7.0. So, these can be the potential drug candidates. The molecular docking result suggested that the ChEMBL ID 358279 is the most suitable drug candidate among these six candidates, with the lowest binding affinity of −8.4. This means that this compound has the strongest interaction with the target protein 7JSU compared to the other bioactive molecules.

## 3. Materials and Methods

The proposed hybrid framework is divided into four modules, as shown in Figure 6. This figure shows that module A has different steps involved in the preparation of the input dataset. However, module B described the development of the QSAR model and the performance comparisons with different state-of-the-art ML models. Module C explains how ADMET analysis is carried out for bioactive molecules. Finally, Module D demonstrates how molecular docking is carried out to validate the results obtained from the ADMET analysis.

### 3.1. Module A: Data Preparation

Module A explains different data preprocessing stages, as follows:

#### 3.1.1. Targeting the Replicating Enzyme

Due to viral enzyme properties, the Main protease, 3C-like protease (3CL^pro^), is a potential drug target among coronavirus proteins. This protease, with papain-like protease (PL^pro^) in the viral RNA, plays a pivotal role in replication and transcription [26]. It is also called the Main protease (M^pro^). It is a highly conserved and replicated key enzyme. Due to these crucial characteristics of 3C-like protease, it is an attractive potential target to investigate the binding inhibitors that can effectively bind the target protein.

#### 3.1.2. Dataset

A dataset of inhibitors (Small Molecules) against SARS coronavirus 3C-like proteinase (Target ID: CHEMBL 3927) is retrieved from the ChEMBL database [27] that is publicly available at https://www.ebi.ac.uk/chembl/ accessed on (accessed on 15 January 2022). The ChEMBL database is manually abstracted from the published literature [27]. It contains bioactive molecules with drug-like features combined with chemical, functional, bioactivity, and genomic data to contribute to the transformation of genomic information into new potent drugs. It contains all information for molecules related to the drug-likeness and few properties of ADMET measures. Currently, this database consists of 18.6 M bioactivity measurements for more than 2.1 M compounds and 14 K protein targets. Information from more than 81,000 research publications has been extracted to develop this database. Our retrieved dataset consists of 133 bioactive molecules out of 8.2 K compounds. These extracted small inhibitory molecules also contain the values of the standard type of IC50.

#### 3.1.3. Data Preprocessing

The description related to the dataset, as explained in the above section, consists of 133 small inhibitory molecules. These bioactive molecules are measured in standard unit IC50 values in nM (nanoMol). The molecules with no IC50 values are dropped. Duplicated data are also deleted. To normalize the IC50 data distribution, we have taken each bioactive compound to its binding affinity to a target protein and converted it into pIC50 (pIC50 = −log10 (IC50)). After cleaning and preprocessing, the dataset consists of 86 small bioactive molecules. Next, bioactive compounds are labeled as either active, inactive, or intermediate classes based on their IC50 values. Active class compounds are those with IC50 values less than or equal to 1000 nM. However, inactive class compounds have IC50 values greater than or equal to 10,000 nM. The compounds with IC50 values between 1001 nM and 9999 nM are labeled as an intermediate class.

### 3.2. Module B: QSAR Modeling

In our in silico study, quantitative structure–activity relationship (QSAR) models are developed to predict the chemical compounds with the best unknown biological activities. QSAR is a mathematical modeling method for predicting the relationships between the structural properties of known chemical compounds and their unknown biological activities. In QSAR modeling, each compound is characterized by its molecular descriptors, and then the model can be used to predict how the change in the structural property causes a change in biological activity [28]. Structural properties refer to physicochemical properties that represent the structure. However, biological activities refer to pharmacokinetic properties.

#### 3.2.1. Exploratory Data Analysis (EDA)

To check the drug-likeness of the bioactive compounds, Lipinski descriptors are calculated. Lipinski, a scientist of Pfizer, described a set of rules-of-thumb to evaluate the drug-likeness of a chemical compound. The rule outlines the molecular characteristics of a drug’s pharmacokinetics—its absorption, distribution, metabolism, and excretion—which represent how well the drug works in the body (“ADME”) [28]. Lipinski’s rule describes that, in general, an orally active drug should not violate more than one condition of the following criteria:The Molecular Weight (MW) should be less than 500 DaltonThe octanol-water partition coefficient (LogP) should be less than 5The hydrogen-bond-donors (NumHDonors) should be less than 5The hydrogen-bond-acceptors (NumHAcceptors) should be less than 10

By examining Lipinski’s rule-of-five descriptors, the chemical space of 3CL inhibitors was navigated to obtain insight into the structure–activity connection. This chemical space analysis may offer vital information about the fundamental characteristics of substances that control their inhibitory properties. Furthermore, exploratory data analyses via Lipinski descriptors are performed. Figure 7 shows the frequency plot of the active and inactive classes. However, Figure 8 depicts a scatter chart comparing MW with LogP. This figure demonstrates that both bioactivity classes are straddling similar chemical regions.

#### 3.2.2. Feature Extraction

The PubChem database accessed on (20 January 2022) [29] is used to extract features from inhibitory molecules. PubChem is one of the largest databases that possess chemical structures and bioactive molecules. The processed balanced dataset contains 76 small inhibitory molecules in SMILES format. This format is cleaned from salt and standardizing tautomer using built-in functions in the PaDEL descriptor [28]. The PaDEL descriptor calculated the PubChem Substructure Fingerprints using the features of RDKit. Table 6 illustrates the description of PubChem substructure fingerprints. It consists of 881 columns in the form of an order list of 0/1 bits. The bit position 0–114 represents the presence of chemical atoms; the bit position 115–262 represents the presence of the described chemical ring system; the bit position 263–326 denotes the simple atom pairs; the bit position 327–415 represents the simple atom nearest neighbors; the bit position 416–459 represents the detailed atom neighborhoods; the bit position 460–712 implies the simple SMARTS patterns; and the bit position 713–880 signifies the complex SMARTS patterns.

#### 3.2.3. Extra Tree Regressor-Based Ensemble Model

The objective of this study is to build regression models that enable the estimation of the continuous response variable (i.e., pIC50), it being a function of predictors (i.e., PubChem fingerprint descriptors). For this purpose, various ML algorithms are developed for QSAR modeling. Due to the higher prediction performance, we select the Extra Tree Regressor (ETR)-based ensemble approach. This model employs a meta-estimator to fit several randomized decision trees and, during training, pick diverse dataset sub-samples. This algorithm avoids over-fitting and provides improved predictive accuracy.

The input dataset, as explained in Section 3.1.3, consists of a total of 86 bioactive molecules, in which 15, 58, and 13 bioactive molecules belong to the active, inactive, and intermediate classes, respectively. In the literature, many useful sampling techniques exist to balance the input dataset [30,31]. The down-sampling deletes some samples from the majority class at the cost of losing useful information. On the other hand, up-sampling balances the dataset by duplicating the samples from the minority class. To balance the dataset, we have made two adjustments. First, we empirically chose a threshold value of 4.50. Those molecules that have pIC50 values ≥ 4.5 have high potency and are retained in the dataset. These molecules may belong to the active, inactive, and intermediate classes. After this adjustment, the 15, 33, and 13 bioactive molecules remain in the active, inactive, and intermediate classes, respectively. In the work, the bioactive molecules belonging to the active class are up-sampled to make 30 molecules. These adjustments made the balanced dataset in both the active and inactive classes. Now, the balanced dataset consists of a total of 76 bioactive molecules. However, the intermediate-class molecules have an equal chance to fit in both active and inactive classes. This balanced dataset is used to train and test the models.

To assess the performance of the regression models, a pair of statistical variables, R^2^ and root mean square error (RMSE), are used. The R^2^ value indicates the fitness of the model. It measures the variance of the dependent variable that is explained by independent variables. Its values range from 0 to 1, highlighting how our data fit the model. The zero value shows that the model does not fit the data, and the one value represents that the model is perfectly fit. RMSE represents the relative error of the predictive model. Finally, a comparative analysis of regression models is carried out.

### 3.3. Module C: ADMET Analysis

The prediction of Absorption, Distribution, Metabolism, Excretion, and Toxicity (ADMET) properties plays a significant role in the drug design process. To evaluate the pharmacokinetics, medicinal chemistry, lipophilicity, water solubility, physicochemical characteristics, and drug-likeness of bioactive compounds, ADMET analysis is performed through the SWISSADME platform [32]. The 3D structure of predictive bioactive compounds is retrieved from PubChem for ADMET analysis. The descriptions related to the CHEMBL ID, Molecular Formula, PubChem ID, Isomeric SMILES, and 3D Structure of these bioactive molecules are given in Table 2. The structure of these chemical compounds is converted into SMILES and fed into the SWISSADME webserver for ADMET analysis. The result of the ADMET analysis decides whether a compound can be a potential drug-like candidate or not. This would help to filter the bioactive molecules for further analysis.

### 3.4. Module D: Molecular Docking

The crystal structure of 3C-like protease (3CLpro) (*PDB ID:* 7JSU) is fetched from the RCSB Protein Data Bank website on 22 January 2022. This structure is purified by removing ligands, water molecules, and alternative side chains. The protein is prepared by adding polar hydrogen atoms and distributing Kollman charges. After making this macromolecule in a charged form, a GridBox is set up to cover the active side of the 7JSU protein. The dimensions of x, y, and z are 30, 30, and 30 with spacing 1, and the centers of x, y, and z are −11.046, 12.826, and 67.789, respectively. Auto-Dock VINA, version 1.2.0; Software for Molecular Docking; The Scripps Research Institute, La Jolla, USA, 2010 [33] with default parameters is used to prepare proteins and ligands to perform molecular docking. The bioactive molecules that are identified after ADMET analysis are used as ligands in molecular docking. After preparing the ligands, molecular docking is performed to calculate the binding affinities (kcal/mol) of these ligands with the target protein 7JSU. The interaction with the lowest binding energy is the best pose. Figure 9 depicts the crystal structure of the SARS-CoV-2 3CL protease 7JSU, with a resolution of 1.83 Å.

AutoDock vina was designed and developed to dock the small drug-like molecules to the proteins with a known structure. The main advantage of this tool is that its performance is evaluated in terms of the diverse types of protein–ligand complexes related to biological and medicinal interest. However, this tool has some limitations as well [34]. For example, it is not suitable for large chemical compounds/ligands. Sometime, the significant conformational flexibility is displayed in the protein targets but not demonstrated in AutoDock vina.

## 4. Conclusions

In this study, we repurposed existing therapeutic agents by examining drug-like bioactive molecules for COVID-19. We developed a hybrid approach that combines useful extracted information through various bioinformatics tools. The Main protease, 3C-like protease (3CL^pro^), is the most suitable potential drug target among coronavirus proteins due to its property as a viral enzyme. The current in silico study has explored the small molecule inhibitors against the infection of COVID-19. The virtual screening was performed on the ChEMBL database and found 133 bioactive molecules against 3CL^pro^. QSAR modeling is developed to predict the chemical compounds with the best biological activities.

Our comparative analysis demonstrated that the proposed Extra Tree Regressor (ETR)-based QSAR model has improved prediction results related to the bioactivity of chemical compounds as compared to Gradient Boosting-, XGBoost-, Support Vector-, Decision Tree-, and Random Forest-based regressor models.

From Lipinski’s rules, we found 86 drug-likeness bioactive molecules against SARS coronavirus 3CL Protease. From the ADMET analysis on the active class data, we identified thirteen novel bioactive molecules for SARS-CoV-2. In the next step, the efficacy of bioactive molecules is computed in terms of binding affinity using molecular docking. This technique has shortlisted the six most suitable bioactive molecules, with the ChEMBL IDs 187460, 222769, 225515, 358279, 363535, and 365134. These molecules can further be investigated as drug candidates for SARS-CoV-2 3CL Protease. The pharmacologist community can adopt these short-listed, relatively small-sized bioactive molecules to develop potential drug candidates.

## Figures and Tables

**Figure 1 pharmaceuticals-15-01333-f001:**
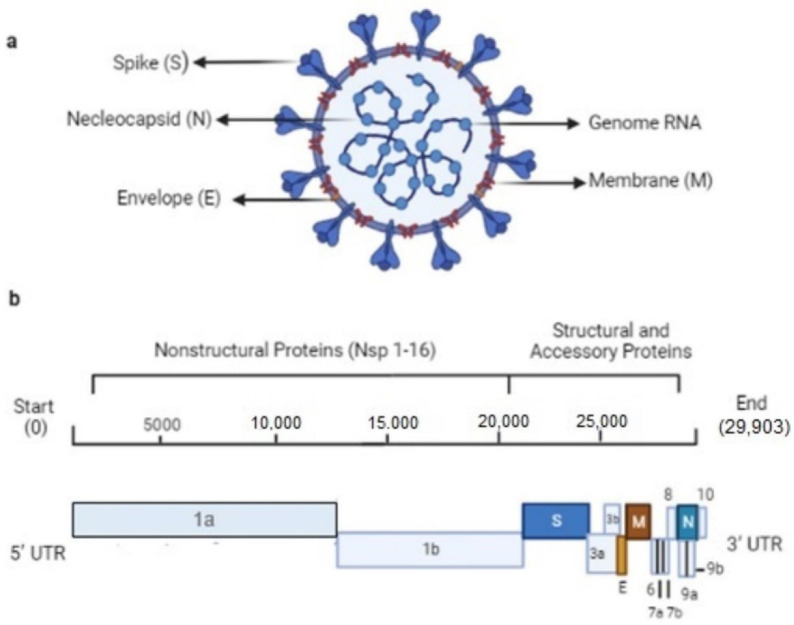
(**a**) SARS-CoV-2 with constituent proteins, (**b**) related genome detailed information.

**Figure 2 pharmaceuticals-15-01333-f002:**
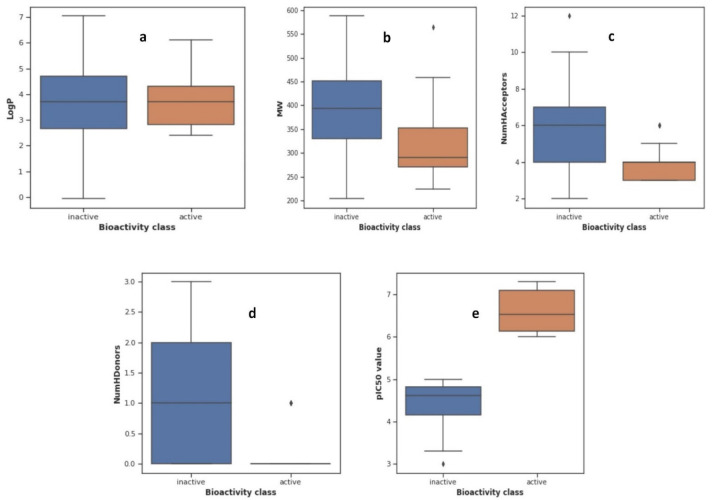
(**a**–**e**): Mann–Whitney U test for LogP, MW, NumHacceptors, NumHdonors, and pIC50.

**Figure 3 pharmaceuticals-15-01333-f003:**
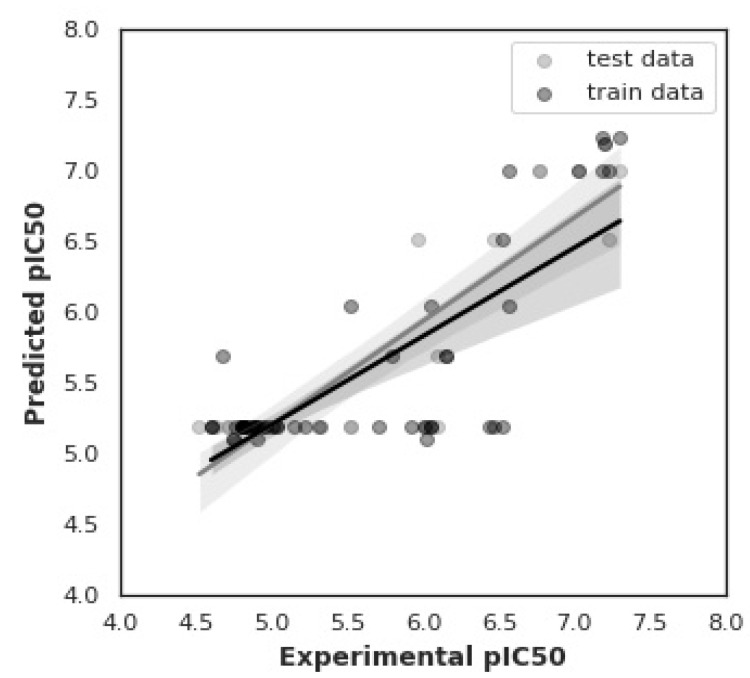
Experimental versus predicted pIC50 using the ETR model for training and testing data.

**Figure 4 pharmaceuticals-15-01333-f004:**
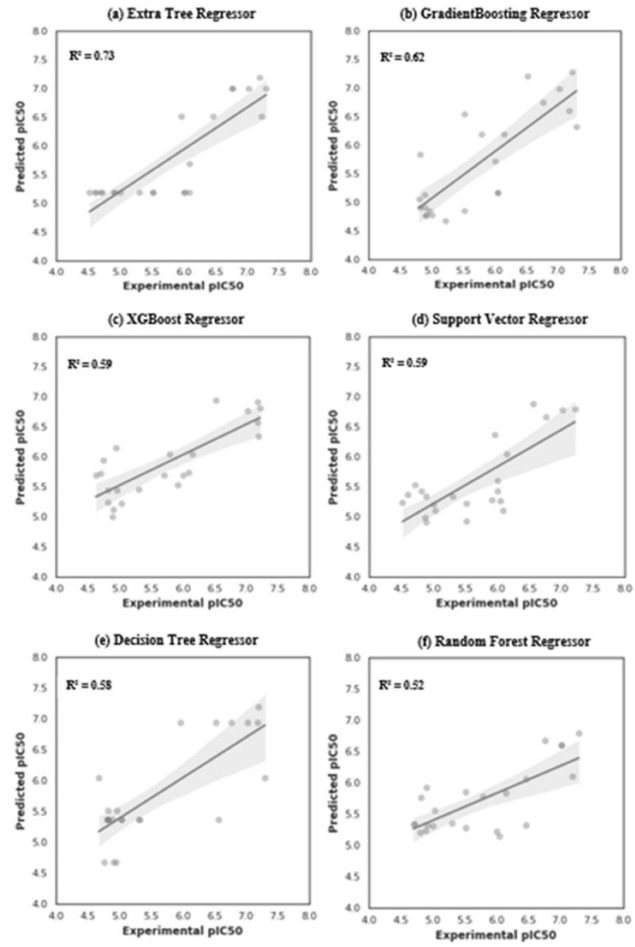
Plots of experimental versus predicted pIC50 for the ETR, GBR, XGBR, SVR, DTR, and RFR models.

**Figure 5 pharmaceuticals-15-01333-f005:**
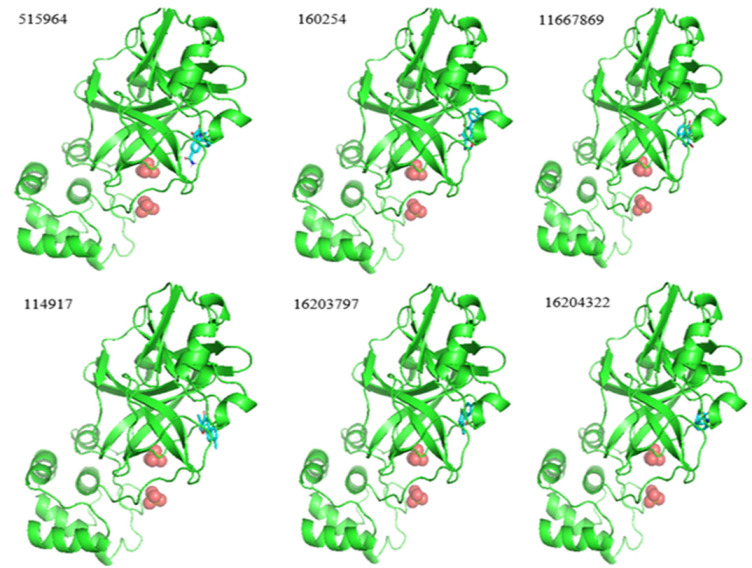
Best pose of six bioactive molecules towards the target protein 7JSU.

**Figure 6 pharmaceuticals-15-01333-f006:**
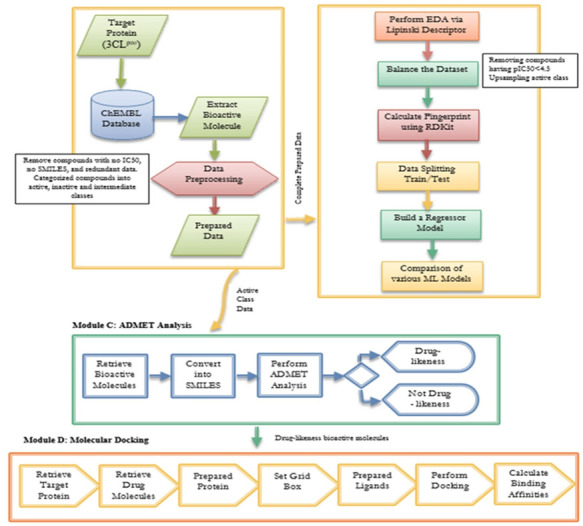
Four main modules (A to D) in the proposed hybrid framework.

**Figure 7 pharmaceuticals-15-01333-f007:**
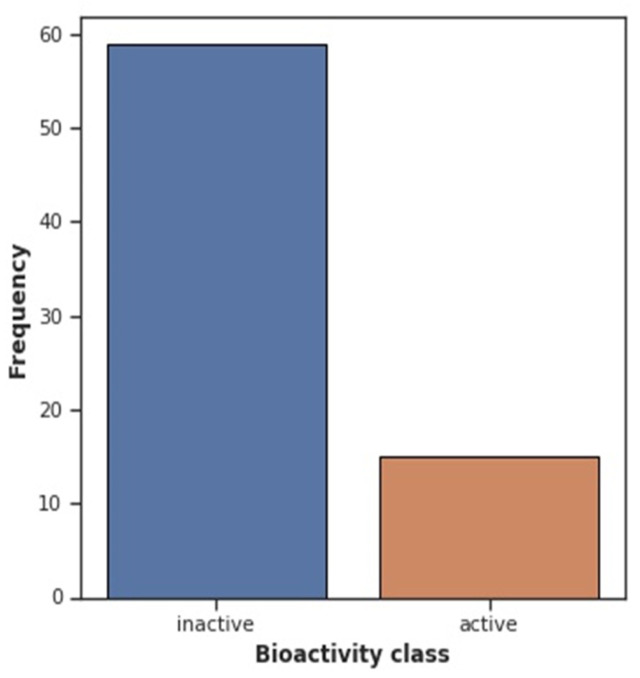
Frequency plot of Bioactivity Classes.

**Figure 8 pharmaceuticals-15-01333-f008:**
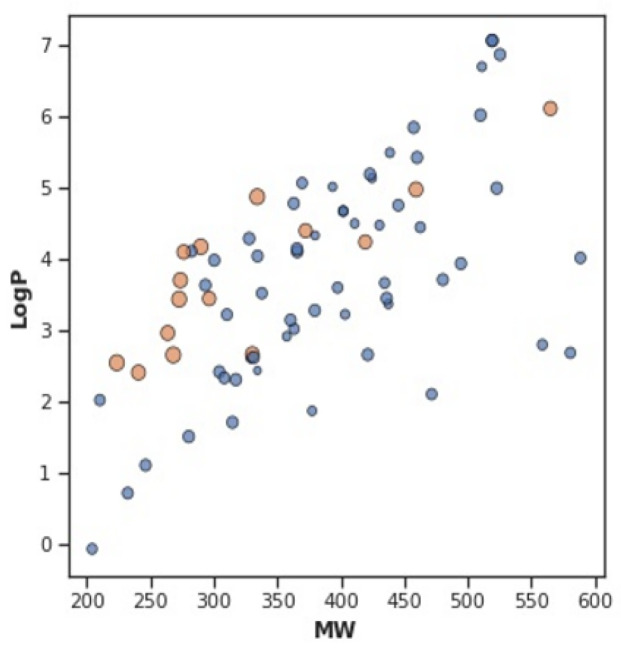
Scatter plot of MW vs. LogP.

**Figure 9 pharmaceuticals-15-01333-f009:**
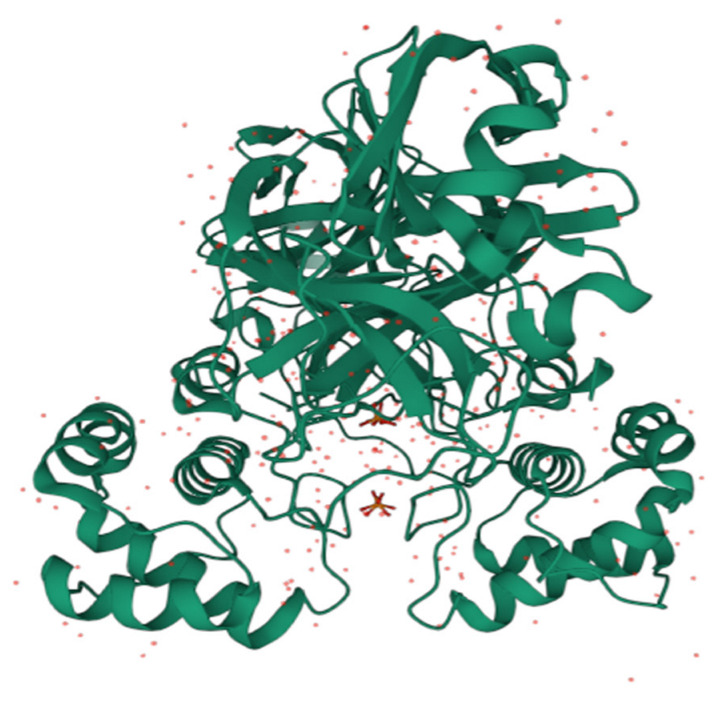
Crystal structure of SARS 3CL protease 7JSU.

**Table 1 pharmaceuticals-15-01333-t001:** Mann–Whitney U test Results.

Descriptor	Statistics	*p*	Alpha	Interpretation
LogP	440	0.4892	0.05	Same distribution (fail to reject H0)
MW	232	0.0023	0.05	Different distribution (reject H0)
NumHAcceptors	214.5	0.0009	0.05	Different distribution (reject H0)
NumHDonors	157	0.00002	0.05	Different distribution (reject H0)
pIC_50_	0	1.37 × 10^−9^	0.05	Different distribution (reject H0)

**Table 2 pharmaceuticals-15-01333-t002:** Description of Fifteen Bioactive Molecules.

CHEMBL ID	Molecular Formula	PubChem ID	Isomeric SMILES	3D Structure
CHEMBL 187460	C_19_H_20_O_3_	160254	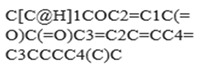	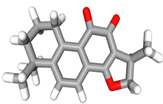
CHEMBL 190743	C_17_H_10_INO_2_S	11796320	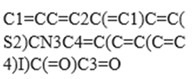	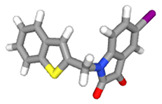
CHEMBL 212218	C_14_H_7_Cl_2_F_3_N_2_O_6_S	2799606	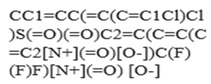	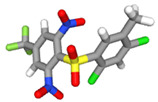
CHEMBL 212454	C_18_H_8_Cl_6_O_6_S	2774892	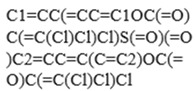	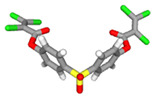
CHEMBL 222234	C_10_H_6_BrNO_3_	16203681	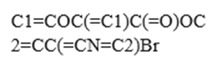	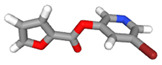
CHEMBL 222628	C_9_H_5_ClN_2_O_2_S	16203796	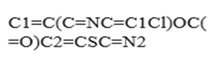	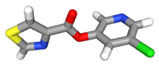
CHEMBL 222735	C_13_H_10_ClNO_3_	16204324	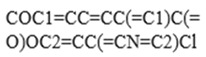	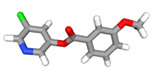
CHEMBL 222769	C_16_H_9_Cl_2_NO_3_	16203797	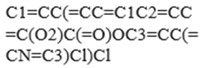	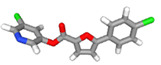
CHEMBL 222840	C_10_H_6_ClNO_3_	7230550	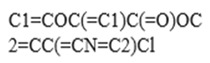	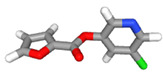
CHEMBL222893	C_14_H_8_ClNO_2_S	2800273	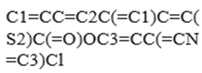	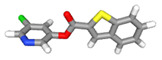
CHEMBL 225515	C_14_H_9_ClN_2_O_2_	16204322	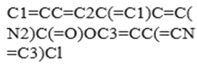	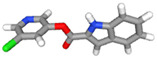
CHEMBL 358279	C_20_H_14_N_2_O_3_	515964	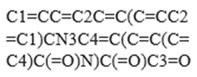	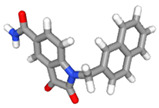
CHEMBL 363535	C_18_H_12_O_3_	114917	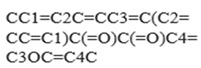	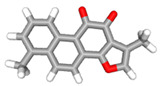
CHEMBL 365134	C_17_H_10_BrNO_2_S	11667869	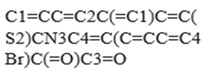	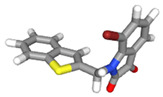
CHEMBL 426898	C_14_H_8_ClNO_3_	16204318	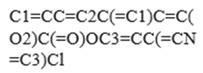	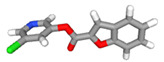

**Table 3 pharmaceuticals-15-01333-t003:** Performance Comparison of Various Regression Models.

Regression Model	R-Squared	MSE	RMSE
Extra Tree Regressor	0.73	0.005	0.074
Gradient Boosting Regressor	0.62	0.006	0.078
XGBoost Regressor	0.59	0.008	0.089
Support Vector Regressor	0.59	0.006	0.078
Decision Tree Regressor	0.58	0.008	0.092
Random Forest Regressor	0.52	0.008	0.089

**Table 4 pharmaceuticals-15-01333-t004:** ADMET Analysis.

ChEMBL ID	Physicochemical Properties	Lipophilicity	Water Solubility	Pharmacokinetics	Drug-Likeness	Medicinal Chemistry
187460	MW 324.58 g/mol TPSA 43.37 Å^2^ NHA = 3 NHD = 0	Consensus log *P*_o/w_ 3.06	Moderately soluble	GI absorption = High BBB Permeant = Yes Skin Permeation (log K_p_) = −5.57cm/s	Yes	Synthetic accessibility = 5.25
190743	MW 442.42 g/mol TPSA 69.47 Å^2^ NHA = 2 NHD = 0	Consensus log *P*_o/w_ 2.43	Moderately soluble	GI absorption = High BBB Permeant = Yes Skin Permeation (log K_p_) = −6.91cm/s	Yes	Synthetic accessibility = 5.36
212218	MW 459.18 g/mol TPSA 136.16 Å^2^ NHA = 9 NHD = 0	Consensus log *P*_o/w_ 3.58	Moderately soluble	GI absorption = Low BBB Permeant = No Skin Permeation (log K_p_) = −5.64cm/s	Yes	Synthetic accessibility = 3.13
212454	MW 585.19 g/mol TPSA 86.04 Å^2^ NHA = 6 NHD = 0	Consensus log *P*_o/w_ 3.73	Poorly soluble	GI absorption = Low BBB Permeant = No Skin Permeation (log K_p_) = −6.33cm/s	No	Synthetic accessibility = 8.41
222234Y	MW 276.13 g/mol TPSA 51.21 Å^2^ NHA = 4 NHD = 0	Consensus log *P*_o/w_ −0.82	Highly soluble	GI absorption = High BBB Permeant = No Skin Permeation (log K_p_) = −10.08 cm/s	Yes	Synthetic accessibility = 5.02
222628	MW 246.71 g/mol TPSA 59.44 Å^2^ NHA = 4 NHD = 0	Consensus log *P*_o/w_ −1.05	Highly soluble	GI absorption = High BBB Permeant = No Skin Permeation (log K_p_) = −9.62 cm/s	Yes	Synthetic accessibility = 4.63
222735Y	MW 280.81 g/mol TPSA 35.53 Å^2^ NHA = 4 NHD = 0	Consensus log *P*_o/w_ 0.73	Very soluble	GI absorption = High BBB Permeant = Yes Skin Permeation (log K_p_) = −8.19cm/s	Yes	Synthetic accessibility = 5.74
222769	MW 350.28 g/mol TPSA 43.37 Å^2^ NHA = 4 NHD = 0	Consensus log *P*_o/w_ 0.58	Very soluble	GI absorption = High BBB Permeant = Yes Skin Permeation (log K_p_) = −9.57cm/s	Yes	Synthetic accessibility = 6.00
222840	MW 231.68 g/mol TPSA 51.21 Å^2^ NHA = 4 NHD = 0	Consensus log *P*_o/w_ −0.90	Highly soluble	GI absorption = High BBB Permeant = No Skin Permeation (log K_p_) = −9.85cm/s	Yes	Synthetic accessibility = 4.74
222893Y	MW 305.86 g/mol TPSA 58.39 Å^2^ NHA = 3 NHD = 0	Consensus log *P*_o/w_ 1.04	Very soluble	GI absorption = High BBB Permeant = Yes Skin Permeation (log K_p_) = −7.90cm/s	Yes	Synthetic accessibility = 5.75
225515Y	MW 287.81 g/mol TPSA 26.30 Å^2^ NHA = 4 NHD = 0	Consensus log *P*_o/w_ 0.06	Very soluble	GI absorption = High BBB Permeant = No Skin Permeation (log K_p_) = −9.53cm/s	Yes	Synthetic accessibility = 5.50
358279	MW 360.57 g/mol TPSA 80.47 Å^2^ NHA = 3 NHD = 1	Consensus log *P*_o/w_ 2.06	Soluble	GI absorption = High BBB Permeant = No Skin Permeation (log K_p_) = −6.45cm/s	Yes	Synthetic accessibility = 4.55
363535Y	MW 301.48 g/mol TPSA 51.21 Å^2^ NHA = 3 NHD = 0	Consensus log *P*_o/w_ 2.29	Soluble	GI absorption = High BBB Permeant = Yes Skin Permeation (log K_p_) = −5.95cm/s	Yes	Synthetic accessibility = 5.35
365134	MW 393.40 g/mol TPSA 69.47 Å^2^ NHA = 2 NHD = 0	Consensus log *P*_o/w_ 2.05	Soluble	GI absorption = High BBB Permeant = Yes Skin Permeation (log K_p_) = −7.30cm/s	Yes	Synthetic accessibility = 6.47
426898Y	MW 289.80 g/mol TPSA 43.37 Å^2^ NHA = 4 NHD = 0	Consensus log *P*_o/w_ 0.57	Very soluble	GI absorption = High BBB Permeant = Yes Skin Permeation (log K_p_) = −8.23cm/s	Yes	Synthetic accessibility = 5.60

**Table 5 pharmaceuticals-15-01333-t005:** Docking results of thirteen selected ligands towards the target protein 7JSU.

Protein Name	ChEMBL ID	Ligand ID	Binding Affinity (kcal/mol)
7JSU	187460	160254	−8.0
7JSU	190743	11796320	−6.7
7JSU	222234	16203681	−5.4
7JSU	222628	16203796	−5.4
7JSU	222735	16204324	−6.6
7JSU	222769	16203797	−7.3
7JSU	222840	7230550	−5.3
7JSU	222893	2800273	−6.5
7JSU	225515	16204322	−7.0
7JSU	358279	515964	−8.4
7JSU	363535	114917	−7.6
7JSU	365134	11667869	−7.8
7JSU	426898	16204318	−6.6

**Table 6 pharmaceuticals-15-01333-t006:** PubChem substructure fingerprints description.

Bit Position	Description
0–114	Presence of chemical atoms
115–262	Presence of the described chemical ring system
263–326	Presence of simple atom pairs
327–415	Presence of simple atoms nearest neighbors
416–459	Presence of detailed atom neighborhoods
460–712	Presence of simple SMARTS patterns
713–880	Presence of complex SMARTS patterns

## Data Availability

The datasets generated during and/or analyzed during the current study are available from the corresponding author on reasonable request.

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
