# Peer review of "Hybrid Approach to Identifying Druglikeness Leading Compounds against COVID-19 3CL Protease"

_pharmaceuticals, 2022, doi:10.3390/ph15111333_

Round 1

Reviewer 1 Report

The manuscript “Hybrid Approach to Identify Druglikeness Leading Compounds against COVID-19 3CL Protease” by Aqeel et al reports developing an in-silico study-based hybrid framework for repurposing drugs for treating COVID-19. The manuscript is well written, and results are well presented and discussed by referring to data reported in literature. I recommend publishing it in its current format.

Author Response

Accepted this MS as it is. We appreciable his general comments related to this study for a Nobel cause

Reviewer 2 Report

In this study, the authors reported the use of various in silico tools (e.g. SwissADME, docking) to identify 6 bioactive compounds against SARS CoV 3CL protease. The authors also proposed a QSAR model that may be used to predict activity of new compounds against the target. While the effort by the authors is exemplary, there are few comments that require authors attention:-

1. Section 2.2.3 (lines 271-274) - the description on how the dataset is balanced is difficult to follow and need to be improved.

2.  Section 2.3 (lines 289 - 290) - how a 3D structure of a compound is retrieved using QSAR modeling?

3. Section 3.1 (lines 350) - how the 15 bioactive molecules were selected from the EDA analysis? Please elaborate.

4. Section 3.2/3.3 - R2 is not Pearson's correlation coefficient, and it can't be negative value. Please use the correct terminology and definition. Also, there is a clear overlap between the information in Table 4 and Figure 9 which makes Figure 9 redundant.

5. Section 3.4 - Drug-likeness and medicinal chemistry (synthetic accessibility) are different from ADMET (though all these parameters can be obtained from SwissADME). Also, it is not clear why the authors need to include this analysis since ChEMBL database is curated after take into consideration all information on ADMET and drug-likeness. 

6. Section 3.5 - the rationale for the docking study is also not clear. Since the compounds are already selected from a pool of drug-like bioactive compounds in the ChemBL database against the SARS CoV 3CL protease target, it is unclear why docking still need to be performed to shortlist the 6 potential compounds. Perhaps, it is better to use docking to predict the binding site and mode of the compounds on the target enzyme? Also, it is rather redundant when the authors mentioned at lines 483-486 that all 13 ligands have zero rmsd values when the authors compared the best pose with the best pose?

Overall, the write-up needs improvement. 

Author Response

In this study, the authors reported the use of various in silico tools (e.g. SwissADME, docking) to identify 6 bioactive compounds against SARS CoV 3CL protease. The authors also proposed a QSAR model that may be used to predict activity of new compounds against the target. While the effort by the authors is exemplary, there are few comments that require authors attention:-

Response: Thanks to the reviewer for appreciable comments. Now, we have revised MS text in the light of useful comments:

  1. Section 2.2.3 (lines 271-274) - the description on how the dataset is balanced is difficult to follow and need to be improved.

Response:  To highlight this point, we have explained in the revise MS as follows. [Page 20]

 “In the literature, many sampling techniques are available to balance the input dataset. The down-sampling deletes some samples from the majority class at the cost of losing the useful information. On the other hand, up-sampling balance the dataset by duplicating the samples from the minority class. To balance the dataset, we have made two adjustments. First, we empirically choose a threshold value of 4.50. Those molecules that have pIC50 values ≥ 4.5, have high potency and are retained in the dataset.  These molecules belong to the active, inactive, and intermediate classes. However, in the work, the molecules belonging to active class are up-sampled.”

  1. Section 2.3 (lines 289 - 290) - how a 3D structure of a compound is retrieved using QSAR modeling?

Response: Thanks to the reviewer for pointing this mistake. Now, we have corrected as

“3D structure of predictive bioactive compounds is retrieved from PubChem for ADMET analysis” [Page 21]

  1. Section 3.1 (lines 350) - how the 15 bioactive molecules were selected from the EDA analysis? Please elaborate.

Response: This useful suggestion has been explained in the revise MS as follows. [Page 6]

Original text, “From this EDA analysis, we found fifteen bioactive molecules. These fifteen bioactive molecules with ChEMBL IDs, chemical formulae, PubChem IDs, Isomeric SMILES, and 3D structures are tabulated in Table 3. For further ADMET analysis, these bioactive molecules belonging to the active class are selected.”

The below text is further added for explanation in the Revise MS as follows.

“First, based on their IC50 values, we categorize whether the bioactive molecule belongs to the active, inactive, or intermediate class. Then, we applied the statistical test of the Mann-Whitney U-test to find the significant difference between active and inactive classes. At the outcome of this analysis, we found fifteen bioactive molecules.”

  1. Section 3.2/3.3 - R2 is not Pearson's correlation coefficient, and it can't be negative value. Please use the correct terminology and definition. Also, there is a clear overlap between the information in Table 4 and Figure 9 which makes Figure 9 redundant.

Response: Thanks for pointing this mistake. In the study, we have reported the results in terms of R2 measure. Actually, the values of R2 and R represent the coefficient of determination of regression model and the Pearson’s correlation coefficient, respectively. Now, we have corrected this mistake in the revised MS as follows: [see page 8 and similar modification at page 20 ].

“The value of R2 highlights the statistical metric of fit that measures how much variation of a dependent variable (pIC50) is explained by the independent features/variables. Its value ranges from [0 -1]. The higher the value of R2, the better the model would be. ”

Figure 9 is the pictorial description of the numerical results presented in Table 4. Now, to minimize the redundancy as suggested, we have removed figure 9.

  1. Section 3.4 - Drug-likeness and medicinal chemistry (synthetic accessibility) are different from ADMET (though all these parameters can be obtained from SwissADME). Also, it is not clear why the authors need to include this analysis since ChEMBL database is curated after take into consideration all information on ADMET and drug-likeness. 

Response: Thanks to the reviewer for this useful comments/query. Now, we have revised MS text in the light of comments:

We agree that the drug-likeness and medicinal chemistry are different from ADMET properties. But they are related to the parameters of ADMET and play a vital role to assess the chance for a molecule to become a useful oral drug. Although, ChEMBL database is curated after take into consideration all information related to the drug-likeness, but this database does not take into account all information about ADMET properties that we have acquired from the SwissADME. On the basis of ADMET analysis, we found 13 bioactive molecules out of 15.  This explanation is available in detail in the revised MS at [11-14] pages.

  1. Section 3.5 - the rationale for the docking study is also not clear. Since the compounds are already selected from a pool of drug-like bioactive compounds in the ChemBL database against the SARS CoV 3CL protease target, it is unclear why docking still need to be performed to shortlist the 6 potential compounds. Perhaps, it is better to use docking to predict the binding site and mode of the compounds on the target enzyme? Also, it is rather redundant when the authors mentioned at lines 483-486 that all 13 ligands have zero rmsd values when the authors compared the best pose with the best pose?

Response: Thanks to the reviewer for useful comments and query.

“In the proposed approach, a total of 133 bioactive compounds are retrieved from a large chemical space at the ChEMBL database against the SARS-CoV 3CL protease target. In order to compute the PubChem fingerprint through RD kit, on the retrieved molecules, we carried out EDA analysis using molecular descriptor with Lipinski's rule. Based on EDA analysis, we determined fifteen biologically active molecule with the drug-like chemical and physical properties.    …………………..……     In the study, from ADMET analysis, we have identified thirteen novel bioactive molecules for SARS-Cov-2. The efficacy of these bioactive molecules is computed in terms of binding affinity using molecular docking and then short-listed the six most favorable potential drug candidates” in the revised MS at [page 3].

Docking technique is also employed to predict the binding site and the compounds mode on the target enzyme as shown in Figure 5. Redundant line has changed at [page 15].

  1. Overall, the write-up needs improvement.

Response: We thoroughly check the revised MS, especially literature review, and efforts are made to improve the quality of write-up.

Round 2

Reviewer 2 Report

The Authors have taken in consideration the comments and suggestions of the reviewers and have attempted to revise the manuscript accordingly. While the paper is much improved, there are still some changes that require the authors' attention before it can be accepted for publication as below:-

1. As there is a rearrangement in the order of the sections, the authors need to recheck the writeup to make sure that the reader can follow based on the new order. For example, in lines 197-198, the authors stated that "The bioactive compounds were categorized as active, inactive, and intermediate classes based on their IC50 value as described earlier". However, this has not been described earlier, as the Methods section is now only mentioned in the later section in the revised manuscript.

2. It is still not clear how EDA analysis outcome feed into QSAR modeling in Module 2 (Figure 6). Rather, it seems that the EDA analysis outcome was used for ADMET analysis instead. The authors need to recheck on this and reorder the results sections accordingly and possibly revise the Modules in Figure 6, accordingly.

3. The quality of the chemical structure in Table 2 needs to be improved.

4. Please change the abbreviation for Topological polar surface area to TPSA (not TSPA). Also, hydrogenbonds should be two words (not one word e.g. "number of hydrogenbonds donor" should be "number of hydrogen bonds donor", etc.).

5. Lines 316-318: penetrate from BBB? or penetrate through BBB?

6. Grammatical errors/hanging sentences are still observed in the manuscript. I suggest the authors to get English language editing service or a native English speaker to go through the manuscript before resubmit.

7. The RMSD values for the docking results is not useful when only one pose is reported. I suggest the authors to remove the RMSD results and the number of best poses, both from text and Table 5. 

8. Lines 491-497: The elaboration of how the dataset is balanced is still difficult to follow. Further improvement is required.

9. The authors should also include the key points from the response below  into the main text accordingly, with proper reference.

 "We agree that the drug-likeness and medicinal chemistry are different from ADMET properties. But they are related to the parameters of ADMET and play a vital role to assess the chance for a molecule to become a useful oral drug. Although, ChEMBL database is curated after take into consideration all information related to the drug-likeness, but this database does not take into account all information about ADMET properties that we have acquired from the SwissADME"

10. The authors should also be more careful with interpretation/conclusion made based on docking result, and should include a sentence or two to highlight its limitation. It has been previously reported that Autodock Vina is more accurate in predicting binding pose, but not so much on the binding affinity. 

Author Response

The Authors have taken in consideration the comments and suggestions of the reviewers and have attempted to revise the manuscript accordingly. While the paper is much improved, there are still some changes that require the authors' attention before it can be accepted for publication as below:-

Response: We thank the reviewer for his appreciable comments. We have revised the MS text in light of suggestions:

  1. As there is a rearrangement in the order of the sections, the authors need to recheck the writeup to make sure that the reader can follow based on the new order. For example, in lines 197-198, the authors stated that "The bioactive compounds were categorized as active, inactive, and intermediate classes based on their IC50 value as described earlier". However, this has not been described earlier, as the Methods section is now only mentioned in the later section in the revised manuscript.

Response: Thank you for pointing this out. We have revised the MS accordingly as follows:

“The bioactive compounds were categorized as active, inactive, and intermediate classes based on their IC50 value. Its detail is given in section 3.1.3.” [Page 4]

  1. It is still not clear how EDA analysis outcome feed into QSAR modeling in Module 2 (Figure 6). Rather, it seems that the EDA analysis outcome was used for ADMET analysis instead. The authors need to recheck on this and reorder the results sections accordingly and possibly revise the Modules in Figure 6, accordingly.

Response: As suggested, we have explained/revised the modules related to Figure 6:

EDA Analysis is used to find the significant difference of both active and inactive classes. For this purpose, Mann-Whitney U test is employed using Lipinski Descriptor. Then we balanced the dataset and fingerprint are calculated, as shown in updated Figure 6 Module B.  

Revised section 3.1.3 include Data preprocessing and categorization in the active, inactive and intermediate classes. The preprocessed data is not only used in QSAR Modeling but also a part of this data (Active Class Data) is used for ADMET analysis as shown in Module A, Module B, and Module C in Figure 6 at [16-17] pages.

  1. The quality of the chemical structure in Table 2 needs to be improved.

Response: We have improved the quality of chemical structures in Table 2 at [6-7] pages.

  1. Please change the abbreviation for Topological polar surface area to TPSA (not TSPA). Also, hydrogenbonds should be two words (not one word e.g. "number of hydrogenbonds donor" should be "number of hydrogen bonds donor", etc.).

Response: Thank you for pointing this out. We have corrected these mistakes in the revised MS pages [10-13].

  1. Lines 316-318: penetrate from BBB? or penetrate through BBB?

Response: We have corrected this mistake in the revised MS at page 10.

  1. Grammatical errors/hanging sentences are still observed in the manuscript. I suggest the authors to get English language editing service or a native English speaker to go through the manuscript before resubmit.

Response: These pointed out suggestions are incorporated. Grammatical errors/hanging sentences are removed to improve the English write up.

  1. The RMSD values for the docking results is not useful when only one pose is reported. I suggest the authors to remove the RMSD results and the number of best poses, both from text and Table 5. 

Response:  As suggested, results related to RMSD values and best-pose have been removed from the text and Table 5 in the revised MS [Pages 14-15].

  1. Lines 491-497: The elaboration of how the dataset is balanced is still difficult to follow. Further improvement is required.

Response: The useful suggestion is incorporated and explained in revised the MS as follows:

“The input dataset, explained in section 3.1.3, consists of a total of 86 bioactive molecules in which 15, 58, and 13 bioactive molecules belong to the active, inactive, and intermediate classes, respectively. ……. After this adjustment, the number of bioactive molecules 15, 33, and 13 remain in the active, inactive, and intermediate classes, respectively. In the work, bioactive molecules belonging to the active class are up-sampled to make 30 molecules. These adjustments made the dataset balanced in both the active and inactive classes. Now the balanced dataset consists of a total of 76 bioactive molecules. However, the intermediate-class molecules have equal chance to fit in both active and inactive classes. This balanced dataset is used to train and test the models.” on [Page 19].

  1. The authors should also include the key points from the response below into the main text accordingly, with proper reference.

 "We agree that the drug-likeness and medicinal chemistry are different from ADMET properties. But they are related to the parameters of ADMET and play a vital role to assess the chance for a molecule to become a useful oral drug. Although, ChEMBL database is curated after take into consideration all information related to the drug-likeness, but this database does not take into account all information about ADMET properties that we have acquired from the SwissADME"

Response: We have incorporated this suggestion in the revised MS:

“The ChEMBL database is manually abstracted from the published literature [27]. ….It contains all information for the molecules related to the drug-likeness and few properties of ADMET measures.” on Page [17]

  1. The authors should also be more careful with interpretation/conclusion made based on docking result, and should include a sentence or two to highlight its limitation. It has been previously reported that Autodock Vina is more accurate in predicting binding pose, but not so much on the binding affinity.

Response.  Many Thanks. We have revised the MS to include the following:

“Autodock vina was designed and developed to dock the small drug-like molecules to the proteins with known-structure. The main advantage of this tool is that its performance is evaluated on the diverse types of protein-ligand complexes related to biological and medicinal interest. However, this tool has some limitations as well [34]. For example, it is not suitable for large chemical compounds/ligands. Sometime, the significant conformational flexibility is displayed in the protein targets but not demonstrated in AutoDock Vina.”  on [Page 20].